# Effect of Physical Separation with Ultrasound Application on Brewers’ Spent Grain to Obtain Powders for Potential Application in Foodstuffs

**DOI:** 10.3390/foods13183000

**Published:** 2024-09-22

**Authors:** Camila Belén Ruíz Suarez, Heidi Laura Schalchli Sáez, Priscilla Siqueira Melo, Carolina de Souza Moreira, Alan Giovanini de Oliveira Sartori, Severino Matias de Alencar, Erick Sigisfredo Scheuermann Salinas

**Affiliations:** 1Undergraduate Program Chemical Civil Engineering, Faculty of Engineering and Sciences, Universidad de La Frontera, Temuco CP 4780000, Chile; c.ruiz06@ufromail.cl; 2Biotechnological Research Center Applied to the Environment (CIBAMA-BIOREN), Universidad de La Frontera, Temuco CP 4780000, Chile; heidi.schalchli@ufrontera.cl; 3Department of Food Science and Technology, Escola Superior de Agricultura Luiz Queiroz (ESALQ), Universidade de São Paulo, Piracicaba CEP 13418-900, Brazil; priscilla.esalq@gmail.com (P.S.M.); moreirasc1@usp.br (C.d.S.M.); alangosartori@usp.br (A.G.d.O.S.); 4Chemical Engineering Department, Universidad de La Frontera, Temuco CP 4780000, Chile; 5Center of Food Biotechnology and Bioseparations (BIOREN), Universidad de La Frontera, Temuco CP 4780000, Chile

**Keywords:** brewers’ spent grain, sonication, convective air-drying, sieving, powder

## Abstract

Brewers’ spent grain (BSG) is the primary by-product of beer production, and its potential use in food products is largely dependent on its processing, given its moisture content of up to 80%. This study aimed to evaluate the effects of physical separation with ultrasound application on the color, total phenolic content (TPC), antioxidant activity, proximate composition, total dietary fibers, and particle size distribution of BSG powders. Wet BSG (W) was subjected to two processes: one without ultrasound (A) and one with ultrasound (B). Both processes included pressing, convective air-drying, sieving, fraction separation (A1 and B1 as coarse with particles ≥ 2.36 mm; A2 and B2 as fine with particles < 2.36 mm), and milling. The total color difference compared to W increased through both processes, ranging from 1.1 (B1 vs. A1) to 5.7 (B1 vs. A2). There was no significant difference in TPC, but process B powders, particularly B2, showed lower antioxidant activity against ABTS•+, likely due to the release of antioxidant compounds into the liquid fraction during pressing after ultrasound treatment. Nonetheless, process B powders exhibited a higher content of soluble dietary fibers. In conclusion, ultrasound application shows potential for further extraction of soluble fibers. However, process A might be more practical for industrial and craft brewers. Further studies on the use of the resulting BSG powders as food ingredients are recommended.

## 1. Introduction

Brewers’ spent grain (BSG) is a plentiful by-product of beer production, primarily composed of the husks of barley malt grain. However, its use in food products is limited due to its high susceptibility to microbial spoilage [1]. It is estimated that more than 180 million tons of BSG are produced globally each year, accounting for 85% of brewery residues, with most of it being discarded or used as low-value animal feed [2]. On a dry basis, BSG is nutritionally rich, containing 50 to 70% dietary fiber—such as arabinoxylans, cellulose, hemicellulose, and lignin—along with 15 to 25% protein, 7 to 10% lipids, 1 to 12% starch, and various vitamins and minerals [3]. BSG is also a valuable source of phenolic compounds, including phenolic acids, flavonoids, tannins, proanthocyanidins, and aminophenolic compounds [4]. However, with a moisture content of 70 to 80%, BSG is prone to rapid microbiological deterioration. Drying is typically employed to stabilize BSG by reducing its moisture level to around 6%, but this process demands considerable energy [3,5,6]. To mitigate the energy consumption associated with drying, pre-pressing the BSG to lower its initial moisture content has been suggested [7].

Traditionally, the separation of commercially valuable components from BSG has been achieved through chemical methods. However, the use of harsh solvents in these processes can lead to the generation of residues and contaminants, presenting environmental challenges in the valorization process [8]. Considering this, a limited number of studies have explored the use of physical methods, such as ultrasound [9], pressing [10], grinding and centrifugation [11], and sieving [12] in the treatment of BSG. Given that BSG is a lignocellulosic by-product, high-power ultrasound can be particularly effective in breaking down cell structures, thereby facilitating the release of nutrients and other compounds contained within the grains. Ultrasound technology, therefore, holds potential for large-scale processes involving the pretreatment of BSG’s lignocellulosic biomass [9].

Hassan et al. [9] evaluated the use of ultrasound pretreatment as a technique to enhance the saccharification of BSG. Similarly, in the study by Tang et al. [13], ultrasound was used to optimize protein extraction from 1 g of BSG in 100 mL of extractant. Optimal conditions—an extraction time of 82.4 min, ultrasonic power of 88.2 W/100 mL, and a solid-to-liquid (S/L) ratio of 2.0 g/100 mL—resulted in a significant improvement in yield.

Ultrasound has also been employed in combination with convective drying to enhance moisture migration and improve the quality of dried products. Applied prior to drying, ultrasound not only improves moisture removal but also enhances the structural integrity, chemical composition, texture, and retention of bioactive compounds [14,15,16].

Milling, a critical unit operation, reduces the molecular size of polysaccharides by disrupting cell structures and decreasing cellulose crystallinity. This reduction in particle size facilitates the hydrolysis of cellulose, hemicellulose, and lignin, while also decreasing the degree of polymerization of these molecules [17]. Additionally, milling can increase the exposure of proteins within the matrix to proteases during digestion, potentially improving protein digestibility. Further research in this area is warranted. Ibbett et al. [11] proposed a novel hydro-mechanical process using a colloid mill to break down residual aleurone and BSG endosperm tissues, resulting in a protein-rich material with particle sizes ranging from 1 to 10 μm. This protein fraction is then separated by centrifugation, yielding a material with a high protein content in small particles.

The ability to obtain different fractions with specific particle sizes tailored to various applications can significantly enhance the usability of BSG in food products. Sieving plays a crucial role in this process by producing homogeneous fractions with specific sizes. Shih et al. [12] used sieving to produce a homogeneous 800 μm fraction of BSG flour, which was then incorporated into muffins to improve their nutritional content without significantly altering their sensory properties.

BSG can be converted into powder form for use as a food ingredient, making it an attractive subject for exploration. Therefore, it is important to investigate BSG-based powders produced through ultrasound treatment combined with the physical separation of stabilized BSG by drying. The objective of this study was to evaluate the effects of physical separation with ultrasound application on the instrumental color, total phenolic content, antioxidant activity, proximate composition, total dietary fibers, soluble and insoluble fibers, and particle size distribution of BSG powders for potential use in food products.

## 2. Materials and Methods

### 2.1. Raw Material

The BSG used in this study was generously provided by a craft brewery, Birrell Ltda., located in Villarrica, Chile, in the Araucanía Region. The BSG was portioned into 1.010 g samples, packaged in Ziploc^®^ bags (26.8 × 27.3 cm, Racine, WI, USA), and immediately frozen at −18 °C. The additional 10 g in each sample was allocated for moisture measurement in each of the experimental replicates after the BSG was thawed. This step was taken to ensure the accuracy of the measurements required for the study. Each bag was thawed in a refrigerator for 24 h at a temperature of 4–5 °C. Once thawed, the BSG underwent a series of operations as outlined in Figure 1.

### 2.2. Process on BSG Solid Fraction 

Two sequences of operations, referred to as Processes A and B, were evaluated in this study (Figure 1). Each process was repeated six times (in sextuplicate), producing powders labeled A1, A2, B1, and B2.

During each process, the moisture content and color of the BSG were assessed at various stages, indicated in Figure 1 as wet original (W), pressed (P), dried (D), first ultrasound (U1), second ultrasound (U2), and powder (A1, A2, B1, B2). Moisture content and instrumental color were measured following the procedures described in the Section 2.3. “Characterization of the BSG Powders”.

**Figure 1 foods-13-03000-f001:**
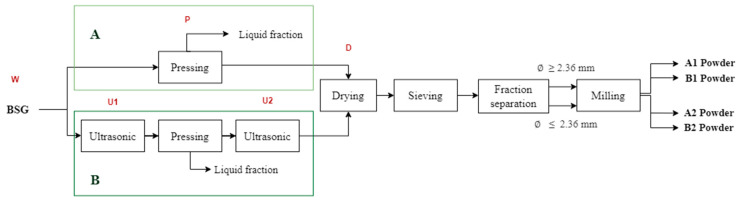
Sequences of operations for processes A and B to obtain food powders from BSG and sampling points. Process A is without ultrasonic application and process B includes two steps of ultrasonic applications. Letters in red indicate the processes within the boxes below them. W is wet original BSG; P is pressed BSG; U1 is the first ultrasound application to BSG; U2 is the second ultrasound application to BSG; D is dried BSG; A1, A2, B1 and B2 are powders of BSG.

#### 2.2.1. Ultrasound Treatment

For the ultrasound operation, an ultrasonic device (NHF-2010, HumanLab Instrument Co., Hwaseong, Republic of Korea) was used at its maximum frequency of 40 kHz for 60 min [9]. Each 500 g sample of BSG was placed in a Ziploc^®^ bag (26.8 × 27.3 cm, USA) and introduced into the ultrasonic equipment during process B (at points U1 and U2 in Figure 1). The bags containing BSG were fully submerged in 3 L of distilled water within the ultrasonic bath, ensuring they did not float during treatment. The water temperature was monitored and recorded every 5 min using the equipment’s display.

#### 2.2.2. Pressing

Pressing was conducted using a manual press equipped with a 2-L stainless steel container and a mechanical screw. In each repetition, 1000 g of BSG was pressed from the wet original (W) in Process A or after the first ultrasound treatment (U1) in Process B. This operation produced two fractions: liquid and solid (Figure 1). The liquid fraction was stored in Schott glass bottles and frozen at −18 °C for preservation, while the solid fraction was subjected to further processing to obtain powders. The yield of the pressing operation, in terms of liquid fraction, was determined by dividing the volume of the liquid fraction by the initial mass of BSG.

#### 2.2.3. Convective Air-Drying

Drying was performed using a universal oven (UFB-500, Memmert, Schwabach, Germany) set at 70 °C with air circulation at a nominal velocity of 2.5 m/s for 3.5 h. A 500 g mass of BSG was evenly spread over a plastic mesh area measuring 32 × 48 cm, with 2 mm perforations to prevent sample loss. The mesh was placed on a tray with 10 mm diameter holes, creating a layer of approximately 1 cm thickness.

#### 2.2.4. Sieving

For the sieving processes in both A and B, U.S.A. Standard Test Sieves Nos. 6, 7, 8, 10, 14, and 18 (Gilson Company Inc., Lewis Center, OH, USA) were used, featuring specific openings of 3.35 mm, 3 mm, 2.36 mm, 2 mm, 1.41 mm, and 1 mm, respectively. A collection container was placed beneath Sieve No. 18. Sieving was conducted using the Ro-Tap equipment (RX-29, W.S. Tyler, Mentor, OH, USA) for 15 min, with shaking and vibration to facilitate separation.

#### 2.2.5. Fraction Separation

Two fractions were obtained by separating the BSG mass retained on the sieves mentioned above, as illustrated in Figure 1. Fractions A1 and B1 (coarse) contained particles with sizes equal to or greater than 2.36 mm, while fractions A2 and B2 (fine) comprised particles smaller than 2.36 mm. The division into these two fractions was made using Sieve No. 8 (2.36 mm), as particles larger than this size were generally whole or agglomerated BSG, while those smaller were split grains.

#### 2.2.6. Milling

To produce powders from fractions 1 and 2 (Figure 1), an ultracentrifugal mill (ZM 200, Retsch, Haan, Germany) equipped with an 80 µm sieve was operated at 12,000 rpm. The resulting powder samples (A1, A2, B1, and B2) were then packed in hermetically sealed aluminum bags and stored at room temperature (20–25 °C) until characterization.

### 2.3. Characterization of the Powders Obtained from BSG

The characterization of the BSG-derived powders involved determining moisture content, instrumental color, total phenolic content, and antioxidant activity against the 2,2′-azinobis-(3-ethylbenzothiazoline-6-sulfonic acid) radical (ABTS•+), proximate analysis, total dietary fiber, soluble and insoluble fibers, and particle size distribution.

#### 2.3.1. Moisture Content

To determine moisture content, ~5 g of BSG were placed in an oven at 105 °C for 2 h and then weighed. The samples were kept at 105 °C until a constant weight was achieved [18,19].

#### 2.3.2. Instrumental Color

The instrumental color of BSG was measured using a Minolta Chromameter CR-200b colorimeter (Tokyo, Japan) in the CIE Lab* color space, as described by Ihl et al. [20]. The instrument was calibrated with a white standard tile (Y = 93.1, x = 0.3140, and y = 0.3212) under CIE condition C (6774 K) illuminant, CIE 1931 2° observer, with a sample thickness of 0.5 cm. The L* value represents lightness, ranging from 0 (black) to 100 (white). The a* value indicates red (+a*) or green (−a*) hues, while the b* value measures yellow (+b*) or blue (−b*) hues.

For color measurement, each BSG sample was evenly distributed in a 9 cm diameter Petri dish. Readings were taken at 12 different points on the surface of each sample.

The total color difference (ΔE) was calculated using Equation (1) and categorized based on the classification by Adekunte et al. [21], where ΔE > 3 indicates very different colors, 1.5 < ΔE < 3 indicates different colors, and ΔE < 1.5 indicates small color differences.
(1)$$\Delta E=\sqrt{{\Delta a*}^{2}+\Delta {b*}^{2}+\Delta {L*}^{2}}$$

The Chroma (C*) was calculated using Equation (2). This quantitative attribute (colorfulness) is used to determine the degree of difference of a hue in comparison to a grey color with the same lightness, with higher values being indicative of higher perceived intensity [22].
(2)$$C^{*}=\sqrt{{a*}^{2}+{b*}^{2}}$$

Hue angle (h*) was determined by Equation (3). It is used to define the difference of a certain color with reference to a grey color with the same lightness. A higher hue angle represents a lesser yellow character in the assays [22].
(3)$$h^{*}={tan}^{-1}\left(\frac{b^{*}}{a^{*}}\right)$$

#### 2.3.3. Hydroalcoholic Extraction 

Hydroethanolic extracts of the BSG samples, labeled A1, A2, B1, and B2, were prepared following the method described by Meneses et al. [23], with slight modifications. Specifically, 10 mL of a 60:40 *v*/*v* ethanol–water solution was added to 0.5 g of dried BSG samples. The extraction process was conducted using ultrasound (UltraCleaner 1400A, Unique, Indaiatuba, SP, Brazil) for 15 min at 45 °C. Following ultrasound treatment, the samples were centrifuged (Eppendorf 5810R, Eppendorf AG, Hamburg, Germany) at 5000× *g* for 15 min, and the supernatant was collected for further analysis.

#### 2.3.4. Total Phenolic Content

Total phenolic compounds were quantified using a spectrophotometric method based on Al-Duais et al. [24], with modifications. In each well of a microplate, 20 µL of either the standard solution (gallic acid) or BSG extracts were mixed with 100 µL of a 10% Folin–Ciocalteu reagent (diluted in water). After a 5-min incubation period, 75 µL of a 7.5% sodium carbonate aqueous solution was added to each well. A blank was prepared by substituting the sample with distilled water. After 40 min of reaction time, the absorbance was measured at 740 nm using a microplate reader (Multiskan EX, Thermo Fisher Scientific, Waltham, MA, USA). A calibration curve was constructed using gallic acid as the standard, with concentrations ranging from 20 to 120 µg/mL.

#### 2.3.5. Antioxidant Activity against ABTS Radical

The antioxidant capacity was assessed using the ABTS•+ free radical method as described by Al-Duais et al. [24], with some modifications. To prepare the ABTS•+ stock solution, a 7 mM ABTS solution was mixed with 2.45 mM potassium persulfate, and the mixture was allowed to stand in the dark at room temperature for 16 h. The ABTS•+ radical solution was then diluted with 75 mM potassium phosphate buffer (pH 7.4) to achieve an absorbance of 0.700 ± 0.01 at 734 nm. In each microplate well, 20 µL of Trolox solution or BSG extracts were combined with 220 µL of the ABTS•+ radical solution and incubated at room temperature in the dark. After 6 min, the absorbance was measured at 734 nm, using the potassium phosphate buffer as a blank. Trolox was used as a standard at concentrations ranging from 12.5 to 200 µM, and results were expressed as µmol Trolox equivalents per gram of BSG powders.

#### 2.3.6. Proximate Composition

The proximate composition analysis was performed according to the AOAC methods [25]. Total lipids (fat) were evaluated by Soxhlet extraction followed by solvent evaporation. Protein content was determined using the Kjeldahl method, with a nitrogen-to-protein conversion factor of 5.83. Ash content was quantified gravimetrically by calcination at 550 ± 25 °C. Total dietary fiber, including both soluble and insoluble fractions, was measured using an enzymatic method based on the protocol by Asp et al. [26], with some modifications. Total carbohydrates were calculated by subtracting the sum of total lipid, protein, dietary fiber, and ash contents from 100%. The results were expressed as a percentage of dry matter (% *w*/*w*) to facilitate comparison among the different powders.

#### 2.3.7. Particle Size Distribution

The particle size distribution of powders obtained from Processes A and B was determined using U.S.A. Standard Test Sieves Nos. 40, 80, 120, 170, 200, and 325 (Gilson Company Inc., USA), with respective openings of 425, 180, 125, 90, 75, and 45 µm. The collection container was placed beneath Sieve No. 325. Sieving was conducted by shaking and vibrating the samples using Ro-Tap equipment (RX-29, W.S. Tyler, Canada) for 15 min.

### 2.4. Statistical Analysis

The data were analyzed using analysis of variance (ANOVA) with a significance level of *p* < 0.05, employing Minitab^®^ Statistical Software Version 21.0.3. (Chicago, IL, USA). When significant differences were detected, Tukey’s honestly significant difference (HSD) and t-Student post-hoc tests were applied. Data for each process were collected in sextuplicate. Color measurements were also conducted in sextuplicate, as described in the “Instrumental Color” section. Total phenolic content, antioxidant activity against the ABTS•+ radical, total dietary fiber, soluble and insoluble fiber, and particle size distribution were measured in triplicate. Proximate composition was performed in duplicate due to sample quantity limitations and the material required for this analysis.

## 3. Results and Discussion

### 3.1. Process Effects on BSG Solid Fraction

#### 3.1.1. Monitoring of Moisture Content and Instrumental Color

Moisture content and instrumental color were monitored throughout Processes A and B to assess any changes in the BSG attributed to the various operations.

The initial moisture content of the wet original BSG (W) used in both Processes A and B was similar, at 72.4% and 71.2%, respectively, with no significant difference between them (*p* > 0.05). Table 1 presents the moisture content (% wet basis, w.b.) monitored at various sampling points during Processes A and B (Figure 1). These values are consistent with previous studies, which reported moisture contents ranging from 70 to 80% for fresh BSG [1,6,12,27], immediately after the brewing process.

Process B includes ultrasound treatment both before (U1) and after (U2) pressing. Ultrasound treatment is widely used to assist in the extraction of phenolic compounds and proteins, enhance saccharification, and modify the structure of BSG and various biological materials [9,13,16,21,28,29,30,31].

The pressing (P) and drying (D) operations are common to both processes, and the resulting moisture content does not significantly differ (*p* > 0.05) between Processes A and B after these steps. The yields from pressing were 47.2 ± 1.8 mL of liquid fraction per 100 g of original wet BSG (W) in Process A and 53.7 ± 2.5 mL of liquid fraction per 100 g of ultrasound-treated BSG (U1) in Process B. El-Shafey et al. [32] proposed dewatering brewer’s spent grain using a membrane filter press, noting that some processing plants use a two-step process to remove water from BSG: pressing (to achieve a material with less than 65% moisture) followed by drying (to reduce moisture content to less than 10%). Pressing resulted in a decrease of 12.6 percentage points in Process A and 12.1 percentage points in Process B in the moisture content of the wet original BSG (W). Drying at 70 °C for 3.5 h (D) stabilized the BSG at moisture levels of 5.2% and 5.4%, respectively (Table 1), which is similar to or slightly higher than the values reported by Fărcaş et al. [3], Naibaho et al. [5], and Thai et al. [6]. A moisture content of 13% (wet basis) has been recommended for BSG to prevent microbial proliferation, especially by *Bacillus cereus* [33]. Maintaining such moisture levels is crucial to prevent the deterioration of BSG, which is a major challenge in utilizing this by-product of the brewing industry [1].

The moisture content of the powders did not significantly differ (*p* > 0.05) from the moisture content of the dried BSG (D) in both Processes A and B (Table 1). However, a slight increase in moisture content was observed in the powders (A1 and B1) obtained from dried BSG with particle sizes equal to or larger than 2.36 mm, and in the powders (A2 and B2) obtained from dried BSG with particle sizes smaller than 2.36 mm. As shown in Figure 1, the operations of sieving, fraction separation, and milling may have exposed the dried BSG to ambient air, potentially leading to a slight increase in moisture content. The moisture content of powders obtained from Fraction 1 (A1 and B1) was slightly higher than that of powders from Fraction 2 (A2 and B2), but these differences were not statistically significant (*p* > 0.05). This behavior aligns with the observation that larger particles generally have a greater water-holding capacity compared to finer particles, as reported in studies on dietary fiber powders from white cabbage leaves, where fiber matrix damage and pore collapse during grinding were noted [34].

The moisture content of BSG powders (Table 1) ranged from 5.9% to 6.6% (wet basis), which is within the range reported by Shih et al. [12], Okpala and Ofoedu [35], and Baiano et al. [36] for BSG flour, which recorded moisture contents of 12.2%, 5.4–5.6%, and 2.9–3.5%, respectively.

Table 2 presents the instrumental color parameters for wet original BSG (W) in Processes A and B. Both Processes A and B show a significant increase (*p* < 0.05) in lightness (L*) from wet original BSG (W) to the powder forms (A1, A2, B1, and B2). These findings are consistent with the results of Ahmed et al. [37], Hejna et al. [38], and Hejna et al. [39], who reported that lightness is the color parameter most significantly affected by particle size, in studies involving water chestnut flour and extruded and ground BSG processed through thermo-mechanical operations in a twin-screw extruder. Hejna et al. [38,39] reported instrumental color ranges with general L* values from 48.17 to 58.32, a* values from 4.32 to 5.88, and b* values from 12.14 to 14.82 for BSG, which differ from the L* and b* values of the wet original BSG (W) shown in Table 2. These differences could be attributed to variations in barley malts and beer types used in the brewing process.

Process A (Table 2), which includes the wet original BSG (W), pressing (P), drying (D), and the resulting powders (A1 and A2), shows significant differences (*p* < 0.05) in L*, a*, ΔE, and h* color parameters. Lightness (L*) increases significantly throughout the process, with higher values observed in A1 and A2. This increase can be attributed to changes in the material’s surface area after milling, where smaller particles exhibit higher lightness, likely due to an increase in specific surface area that enhances light reflection [38,40,41].

The a* values (Table 2) decrease from P to D, indicating either a reduction in red intensity or an increase in green, and remain stable after sieving and milling (Figure 1) to obtain A1 and A2. The change in a* during the drying phase (D) of BSG could be due to chemical reactions such as the Maillard reaction and caramelization of reducing sugars present in the by-product [12].

The total color difference (ΔE), in comparison to W, progressively increases, with the most significant changes occurring after drying, sieving, and milling to produce the powders. According to Adekunte et al.’s classification [21], the color differences observed relative to the wet original BSG range from very different colors (ΔE > 3) for D, A1, and A2 to a small color difference (ΔE < 1.5) for P.

Regarding h* (Table 2), the BSG samples in the wet original (W), pressed (P), and powder A2 states displayed a significantly lower (*p* < 0.05) yellow character than the BSG samples after drying (D) and in powder A1 [22]. This behavior of h* in Process A could be influenced by two factors. First, the smaller particle size in Fraction 1 compared to Fraction 2, which is associated with an increased specific surface area [38,40,41], and second, the chemical reactions leading to browning, such as the interaction with reducing sugars [12].

The color changes observed in BSG during drying (D) using hot-air methods are likely due to the Maillard reaction and caramelization, which tend to produce a darker product [12]. Additionally, the Maillard reactions in BSG have been documented during extrusion grinding via thermo-mechanical treatment in a twin-screw extruder [38,39].

Process B (Table 2), which includes ultrasound treatments before (U1) and after (U2) pressing, exhibits similar patterns to Process A with significant differences (*p* < 0.05) in L*, a*, ΔE, and h* color parameters. Lightness (L*) is strongly influenced by the drying (D), sieving, and milling operations (Figure 1), with higher values observed for powders B1 and B2. Additionally, Process B shows variations in a* after these treatments, indicating an impact on red/green intensity relative to W. The total color difference (ΔE) increases more significantly after pressing (P) in Process B compared to Process A. However, as in Process A, the most substantial changes in ΔE are observed after drying, sieving, and milling. According to the classification by Adekunte et al. [21], the color changes relative to W range from very different colors (ΔE > 3) for D, B1, and B2, to different colors (1.5 < ΔE < 3) for P and U2, and include small color differences (ΔE < 1.5) for U1. Process B significantly affects (*p* < 0.05) the h* values, with the change occurring during the pressing (P) operation, whereas in Process A, the change occurs during drying (D). As mentioned earlier, the color changes observed in Process B could be attributed to chemical modifications such as the Maillard reaction and caramelization [12,38,39], as well as to the decrease in particle size [38,40,41] primarily caused by drying and milling.

For both processes, the values of b* and C* (chroma) remain unaffected after treatment of W, as no significant differences (*p* > 0.05) were observed. This behavior is consistent with reports by Shih et al. [12], Hejna et al. [38], and Hejna et al. [39], which noted that drying and extrusion at different temperatures did not significantly affect C* from fresh to dried or extruded BSG.

#### 3.1.2. Ultrasonic Temperature Monitoring

Figure 2 depicts the temperature increase in the ultrasonic water bath during U1 and U2 operations, which were applied for 60 min to wet original (W) and pressed (P) BSG, respectively. These graphs are crucial for determining whether the BSG was exposed to temperatures that could induce biochemical changes during ultrasound treatment. The temperature increase can be attributed to the energy release that occurs through acoustic cavitation during ultrasonic operations. Acoustic energy is generated by the transmission of ultrasound waves, which involve cycles of rarefaction and compression traveling through the liquid medium. When cavities formed in the liquid collapse, small amounts of energy are released as heat. The cavitation effects generate high temperatures, pressure, and violent shear forces, leading to the formation of what are referred to as “hot spots” [28]. The final collapse phase is adiabatic, producing locally high-temperature and high-pressure conditions [42].

The slope of the temperature–time curves shown in Figure 2, representing the rate of water temperature increase during U1 and U2, were 0.62 ± 0.04 °C/min and 0.57 ± 0.05 °C/min, respectively. The maximum water temperatures reached during the ultrasound treatment were 55.8 ± 1.9 °C for U1 and 53.8 ± 1.2 °C for U2. No significant differences (*p* < 0.05) were observed in the slope or maximum water temperatures between U1 and U2.

The increase in water temperature within the ultrasonic equipment likely caused a rise in the temperature of the BSG, though this was not directly measured due to the difficulty of inserting a thermocouple into the Ziploc^®^ bag without allowing water infiltration. However, it is estimated that the temperature of the BSG during U1 and U2 did not exceed 55.8 °C and 53.8 °C, respectively. Thus, it is considered that no significant chemical or structural changes occurred in the BSG due to the water temperature increase during the ultrasound operation. Hassan et al. [9] applied ultrasound at temperatures ranging from 20 to 60 °C for 20 to 60 min, and after optimizing the pretreatment of native BSG and subsequent saccharification, 74% of sugars in BSG were recovered, with no lignin degradation observed. In contrast, Alonso-Riaño et al. [29], in their efforts to extract polyphenol compounds from BSG, maintained the temperature at 47 °C during ultrasound-assisted extraction for 30 min (total experiment time of 60 min) to prevent the degradation of these bioactive compounds.

#### 3.1.3. Particle Sieve Size Distribution

Figure 3 shows the particle sieve size distribution, expressed as a percentage of the total mass, for BSG processed through Processes A and B, following sieving. Statistically significant differences (*p* < 0.05) were observed between Processes A and B for the particle size fractions retained on U.S.A. Standard Test Sieves No. 6, 8, 10, 14, and 18 (Gilson Company Inc., USA), with respective openings of 3.35 mm, 2.36 mm, 2 mm, 1.41 mm, and 1 mm. This suggests that ultrasound treatment applied to wet original and pressed BSG in Process B impacts the size distribution of the particles after drying and sieving.

In Process A (Figure 3), the particle sieve size fractions retained on the 3.35-mm and 3.00-mm sieves predominated, accounting for 50.9% (±3.6%) of the total BSG mass after sieving. In contrast, these two particle size fractions represented 43.6% (±3.5%) of the total BSG mass in Process B. The particle sieve size fractions retained on sieves with openings equal to or smaller than 2.36 mm were higher in Process B (56.4% ± 3.0%) compared to Process A (49.1% ± 3.3%). The ultrasound treatment applied in U1 and U2 (Process B) seemingly caused parts of the BSG to break or detach, resulting in structural alterations that led to an increase in the proportion of smaller-sized particles in Process B compared to Process A. This phenomenon may be associated with the effects of ultrasound, which induces cavitation, where microbubbles form, grow, and then collapse violently upon reaching a critical size, converting sonic energy into mechanical energy [9]. This energy could be transmitted to the plant material within the Ziploc^®^ bag, causing the husk, pericarp, and seeds of the BSG grains to break or detach. Bundhoo and Mohee [28] described how ultrasound impacts the physical characteristics of biomass or waste materials by reducing particle size and increasing surface area. Li et al. [31] suggested that ultrasound-assisted extraction exposes more buried SH bonds on the surface by reducing BSG particle size and modifying protein structures.

### 3.2. Characterization of the BSG Powders

As previously mentioned, the BSG powders A1, A2, B1, and B2 from Processes A and B (Figure 1) were obtained by sieving dried BSG into two fractions (1 and 2). These final BSG powders, derived from both operational sequences—Process A (without ultrasound application) and Process B (with ultrasound application)—with different particle size fractions (1 and 2), were then characterized.

#### 3.2.1. Color

Table 3 presents the differences in color (ΔE) between the BSG powders obtained through Processes A and B (Figure 1).

As shown in Table 3, the largest total color difference (*p* < 0.05) was observed between powders A2 and B1. This difference may be influenced by the use of ultrasound (U1 and U2) in Process B, combined with fraction separation (1 and 2), as illustrated in Figure 1 for both processes.

According to the classification by Adekunte et al. [21], very different colors (ΔE > 3) were observed between powders A1 and A2 (Table 3), suggesting that the color difference could be attributed to the structural composition of the two fractions of dried BSG obtained after sieving, which separates particles larger than 2.36 mm (Fraction 1) from those smaller than 2.36 mm (Fraction 2). The powders B1 and B2 showed smaller differences in color (1.5 < ΔE < 3) between each other, following the aforementioned classification. It is proposed that the structural modifications in particle size distribution (Figure 3) produced by ultrasound (U1 and U2) in Process B resulted in powders with smaller color differences compared to Process A.

When comparing powders obtained from the same fractions (1 and 2, above and below 2.36 mm, respectively) but through different processes, the comparison between A1 and B1 (Table 3) showed smaller color differences (ΔE < 1.5) than between A2 and B2 (1.5 < ΔE < 3), according to the Adekunte et al. [21] classification. Our findings suggest that the color changes in the powders due to the ultrasound treatments (U1 and U2) applied in Process B, but not in Process A, affected Fraction 2 (particles smaller than 2.36 mm) more significantly. It is likely that the chemical composition of Fraction 2 particles was more susceptible to the effects of ultrasound than Fraction 1 particles. Ultrasound has been reported to induce various chemical changes in biomass or waste material, as described by Bundhoo and Mohee [28], including alterations in lignin, hemicellulose, cellulose content, cellulose crystallinity, polymerization extent, and organic matter solubilization. It is possible that the particles in Fraction 2, subjected to ultrasound, were more affected by the subsequent operations in Process B than those in Fraction 1. As a result, the impact on color was greater in Fraction 2, likely due to the Maillard reaction or caramelization occurring during the drying step [12,38,39] and the effect of decreased particle size caused by milling [39,40,41] in Process B.

#### 3.2.2. Total Phenolic Content and Antioxidant Activity against ABTS•+ Radical

Table 4 presents the total phenolic content (TPC) and antioxidant activity against the ABTS•+ radical in BSG powders (A1, A2, B1, B2) obtained through Processes A and B.

The application of ultrasound and variations in particle size did not significantly influence the extraction of phenolic compounds from BSG, as indicated by the lack of significant differences in total phenolic content between the treatments (Table 4).

Regarding the total phenolic content (TPC) of BSG, our findings align with previous results reported by Carciochi et al. [43], who found TPC values ranging from 1.59 to 3.57 mg GAE g^−1^ BSG d.m., as well as with the results reported by Alonso-Riaño et al. [29], Petrón et al. [44], and Patrignani et al. [45]. Other studies have reported higher TPC values for BSG extracts, such as those found by Meneses et al. [23] and Bonifácio-Lopes et al. [46], which were 7.13 and 13 mg GAE g^−1^ BSG d.m., respectively. It is important to note that in this study, it was analyzed only the free phenolic compounds, as no hydrolysis step was employed during the extraction process. Since a portion of the phenolic acids present in BSG are bound to the cell wall of lignins, they can be more readily released during alkaline extraction, where the solubility of lignin is enhanced under such conditions [29].

The ABTS assay is a widely used method for evaluating antioxidant activity in vitro, measuring the capacity of antioxidants to neutralize ABTS•+ radicals through electron donation, relative to a standard such as Trolox [24]. The results obtained using the ABTS method showed that antioxidant activity slightly increased in treatments without ultrasound application. This may be explained by the release of antioxidant substances, such as phenolic compounds, capable of scavenging the ABTS•+ radical from the solid to the liquid fraction during Process B, where ultrasound was employed. The initial application of ultrasound likely facilitated the extraction of these ABTS•+ radical scavengers retained in the BSG, which were possibly lost to the liquid fraction during subsequent pressing in Process B (Figure 1). Indeed, the application of ultrasound has been shown to increase the extraction of phenolic compounds from BSG when using water and ethanol as solvents [30]. Additionally, no significant differences (*p* > 0.05) were observed between the different particle size fractions (A1 and A2; B1 and B2) within the processes for antioxidant activity, as measured by the capacity to scavenge ABTS•+ radicals.

#### 3.2.3. Proximate Composition

Table 5 shows the proximate composition (in dry matter, % *w*/*w*) of BSG powders obtained from Processes A and B. The results indicate significant differences (*p* < 0.05) in total dietary fiber, soluble and insoluble fiber, and available carbohydrates among the four types of powder (A1, A2, B1, and B2). However, total lipid, protein, and ash contents did not differ statistically (*p* > 0.05) among them. Lynch et al. [1] provide an extensive compilation of various authors’ reports on the chemical composition of BSG, based on dry matter, noting lipid content of 3–13%, protein content of 14.2–31%, ash content of 1.1–4.2%, hemicellulose content of 19.2–41.9%, cellulose content of 12–33%, starch content of 1–12%, and lignin content of 11.5–27.8%. Our findings for total lipids, protein, total dietary fiber, and ash are within the ranges reported in these studies. Similarly, Shih et al. [12] produced BSG flours using two drying methods to prepare muffins, reporting chemical compositions of 7.1–12.4 g 100 g^−1^ d.m. for lipids, 14.5–18 g 100 g^−1^ d.m. for protein, 2.7–4.1 g 100 g^−1^ d.m. for ash, 8.1–15.9 g 100 g^−1^ d.m. for soluble sugar, and 29.2–49.7 g 100 g^−1^ d.m. for dietary fiber. More recently, Garrett et al. [27] reported the proximate composition of BSG used to prepare flour by drying (65 °C for 72 h) and grinding, with dry basis contents of 4.9% for fat, 16.8% for protein, 2.9% for ash, and 75.4% for carbohydrates.

Total lipid, protein, and ash content in the powders were not significantly affected by the differences between Processes A and B, whether due to the application of ultrasound or the fraction separation methods used, as shown in Figure 1. It is believed that further improvements will be necessary in future studies, particularly concerning the application of ultrasound treatment and sieving, to achieve noticeable differences, especially in protein content. The intensity of cell wall component breakdown and the consequent release of protein during ultrasound treatment of BSG could influence these differences [13,31]. Kissell and Prentice [47] found that in brewers’ spent grain, the finer fractions had a higher protein concentration compared to the coarser fractions, suggesting that the smaller particles may originate from regions of the grain with higher protein concentrations.

The choice of sieve used for fraction separation contributed to differences in the fiber and available carbohydrate content of the powders. Finer fractions (Fraction 2: particles < 2.36 mm) retained more dietary fibers in treatments without ultrasound (Process A). The use of ultrasound, combined with the finer fractions, may be an effective method for increasing the efficiency of dietary fiber extraction from BSG.

According to the study by Hassan et al. [9], the application of ultrasound can increase the concentration of reducing sugars. In our study, for the same fraction, the powders produced without ultrasound (Process A) showed a higher (*p* < 0.05) percentage of available carbohydrates compared to the powders obtained using Process B (Table 5). This behavior could be explained by the release of carbohydrates from the solid to the liquid fraction, promoted by the ultrasonic treatment and subsequent pressing applied after U1 (Figure 1). As a result, more sugar was transferred to the liquid fraction, and less sugar was retained in the solid BSG fraction obtained in Process B.

When considering either of the processes (A or B), it can be observed (Table 5) that fraction separation (Figure 1) into coarse (Fraction 1: equal to or larger than 2.36 mm) and fine (Fraction 2: smaller than 2.36 mm) particle sizes affects the proximate composition of total dietary, soluble, and insoluble fiber, as well as available carbohydrates in the BSG powders obtained in our study. This behavior can be attributed to the fact that Fraction 1 consists of whole grains and agglomerates, while Fraction 2 consists of broken and/or segmented grains. BSG consists of layers of husk, pericarp, and seeds with residual amounts of barley endosperm and aleurone [48]. It is a lignocellulosic material, and its main components are dietary fiber (50%) and protein (30%) [1]. Therefore, sieving and fraction separation into coarse and fine particle sizes of dried BSG (Figure 1) before milling would allow for the production of powders with different proximate compositions. These powders could be applied as specific ingredients in food products, depending on the requirements for higher levels of dietary fiber (Fraction 2) or carbohydrates (Fraction 1).

Table 5 also shows the soluble and insoluble dietary fibers (% *w*/*w*) of BSG powders obtained through Processes A and B. Dietary fibers can represent up to 70% (*w*/*w*) of BSG [49] and are classified according to their solubility in water. Soluble dietary fibers include β-glucans, pectic polysaccharides, arabinogalactans, highly branched arabinoxylans, and xyloglucans, while insoluble dietary fibers include cellulose, hemicellulose, and lignin [48,50]. Barley (*Hordeum vulgare*) and its by-products, such as BSG, are notable as sustainable sources of dietary fibers, offering a mix of soluble and insoluble fibers with numerous health benefits [51], thus allowing for the development of fiber-enriched food products.

Soluble fibers have prebiotic activity, selectively enhancing the growth and/or activity of beneficial bacteria in the colon, thereby contributing to the improvement of intestinal flora and overall host health. Additionally, soluble fibers form colloidal solutions in the intestine, contributing to the reduction in blood sugar and cholesterol levels and aiding in the prevention of cardiovascular diseases, type 2 diabetes, cancer, obesity, and other conditions [52,53]. Insoluble fibers, on the other hand, help in the formation of fecal bolus, accelerate intestinal transit, and play a crucial role in the body’s detoxification [53]. However, they are not fermentable by intestinal flora, which reduces their potential as a functional ingredient. Food processing techniques can be used to modify the structural properties and, consequently, the functional and nutritional roles of insoluble fibers [54].

There were significant differences (*p* < 0.05) in soluble dietary fiber content between the two groups of powders (A and B). The percentage of soluble dietary fiber in powders B1 and B2 exceeded that in A1 and A2, indicating that ultrasound positively affected the soluble fiber content of BSG. Ultrasound treatment has been used for structural modification of BSG and other biological materials [9,16,28,31]. β-glucan can be extracted from BSG and used as a food supplement. Recently, it was demonstrated that BSG β-glucan improved the color and increased the stability of orange juice by reducing pulp precipitation and exhibited higher lightness and color values [55].

Table 5 also shows that both Processes A and B, especially the application of ultrasound and the particle size (Fractions 1 and 2, above and below 2.36 mm, respectively), directly influenced the content of insoluble dietary fiber (*p* < 0.05). Cellulose is a common insoluble dietary fiber found in barley cell walls, as is hemicellulose, which is composed of various sugar monomers, including pentoses like xylose and arabinose, and hexoses like glucose and mannose [51]. According to Lynch et al. [1], hemicelluloses, composed mainly of arabinoxylan, are the main constituent of BSG (up to 40% dry weight), followed by cellulose.

Several factors affect the proximate composition of powder or flour obtained from BSG, including the ingredients used in the beer formulation and the brewing process itself [1,12,27]. Jin et al. [56] presented a detailed proximate analysis of various craft BSGs, comparing them with those from industrial breweries. It was observed that craft BSG has significant differences in its proximate components compared to industrial BSG. Therefore, based on the proximate analysis, it may be possible to determine the origin of the BSG, whether craft or industrial, and further studies should be conducted with different BSG samples to confirm our findings.

#### 3.2.4. Powder Particles Sieve Size Distribution

Table 6 shows the particle sieve size distribution of BSG powders obtained by Processes A and B after milling (Figure 1) using an ultracentrifugal mill equipped with an 80 µm sieve at 12,000 rpm.

In Table 6, the effects of ultrasound applied in Process B and the fraction separation (Figure 1) applied in both processes are evident with respect to the different particle size distributions of the powders. For A1, the highest proportion of particles (36.3%) falls between 90 and 75 µm, for A2 (31.8%) between 75 and 45 µm, for B1 (38.3%) between 125 and 90 µm, and for B2 (27.9%) between 180 and 125 µm. In both A1 (93.8%) and B1 (92.4%), the particles are primarily distributed between 125 and 75 µm, while in A2 (98.4%) and B2 (96.4%), the particles are mainly distributed between 125 and 45 µm. This indicates that the separation of dried (D) and sieved (S) BSG into two fractions (coarse and fine) before milling (Figure 1) had a significant effect on the particle size distribution.

Statistically significant differences (*p* < 0.05) were observed in the percentage of particles retained in the sieves with nominal sizes of 125, 90, 75, and 45 µm. For these three sizes (90, 75, and 45 µm), the fraction separation (Figure 1) into coarse (1: equal to or larger than 2.36 mm) and fine (2: smaller than 2.36 mm) particles affected the percentage of particles between A1 and A2 or B1 and B2. Process B, which included the ultrasound operation, produced a higher proportion (*p* < 0.05) of particles in the 125 and 75 µm range for Fraction 2 compared to Process A. However, for the 90 and 45 µm sizes, A2 showed a higher proportion (*p* < 0.05) of particles than B2. When analyzing Fraction 1, B1 had a higher proportion (*p* < 0.05) of particles at 90 µm than A1, whereas A1 had a higher proportion (*p* < 0.05) of particles at 75 µm than B1.

It was used an 80 µm sieve to mill Fractions 1 and 2 of the four powders obtained. Particles larger than this sieve size were found in all powders, indicating that the milling equipment used in this study was not fully efficient in reducing the particle size below 80 µm. However, particles below 125 µm are considered suitable for many food applications [5,10,12,35,36].

## 4. Conclusions

The two processes developed in this work yielded powders with different characteristics in terms of instrumental color, with ΔE values ranging from 1.1 to 5.7, antioxidant capacity (higher for A1 and A2, as measured by the ABTS assay), proximate composition (mainly with respect to dietary fibers, particularly insoluble fibers which ranged from 33.8% for A1 to 50.7% for B2) and particle size distribution, with statistically significant differences (*p* < 0.05) between A1 and A2 or B1 and B2 for sieves with nominal sizes of 90, 75, and 45 µm.

Process A, without ultrasound treatment, yielded powders with higher antioxidant capacity and is feasible for both industrial and craft brewers, as the operations and equipment required are well-known and could be readily implemented. This process produced differentiated powders that could be used in various applications and products, such as cookies, bread, snacks, dough, bakery products, hamburgers and fruit beverages within the food industry.

On the other hand, Process B, which involves ultrasound treatment, significantly increased the soluble fiber content in the powders, a key attribute for products aimed at fiber-enriched foods. However, the use of ultrasound requires further optimization to fully capitalize on its beneficial effects, particularly its ability to increase soluble fiber content, for application by industrial and craft brewers.

Overall, the powders derived from brewers’ spent grains have a high content of dietary soluble fiber, making them valuable for enriching food products. Additionally, they contribute to protein content and antioxidant activity. Both processes resulted in protein-rich powders, potentially suitable for use in plant-based foodstuffs, which is particularly relevant given the growing vegetarian and vegan population.

Processing BSG for use in the food industry represents a step toward a circular bioeconomy and adds value to this by-product, enhancing food nutrition, especially in regions facing food security challenges. Further studies on the sensory and nutritional effects of using BSG as a food ingredient are encouraged.

## Figures and Tables

**Figure 2 foods-13-03000-f002:**
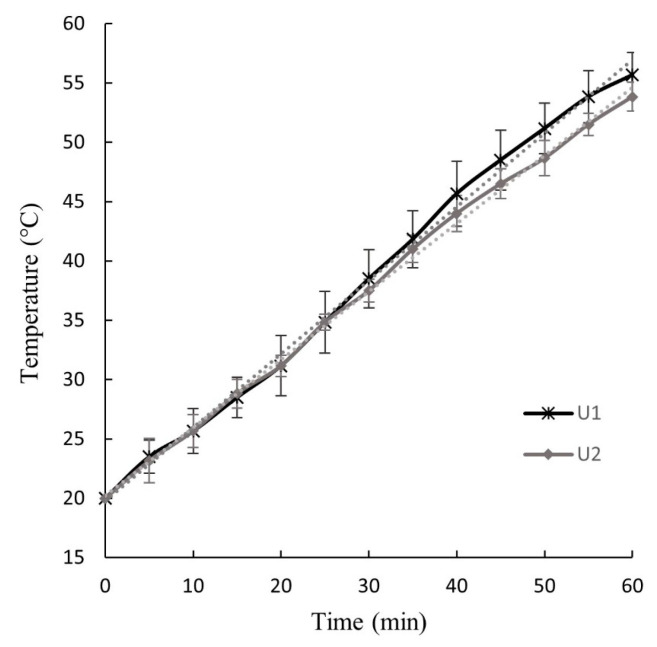
Temperature–time curves of the U1 and U2 ultrasound operation applied to wet original (W) and pressed (P) BSG, respectively.

**Figure 3 foods-13-03000-f003:**
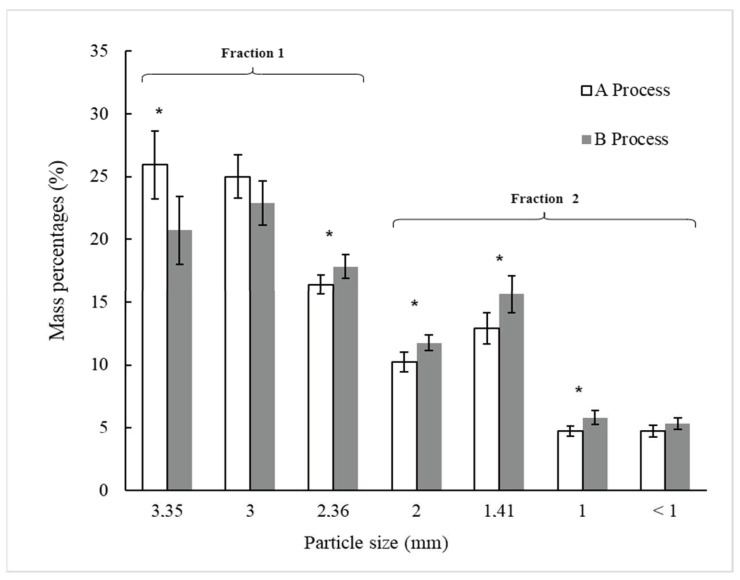
Distribution of particle sieve sizes (in percentage) of dried BSG (D) after sieving from Processes A and B, as outlined in Figure 1. Process A is without ultrasonic application and process B includes two steps of ultrasonic applications. For each particle sieve size, (*) indicates a significant difference between the two processes as determined by the t-Student test at *p* < 0.05.

**Table 1 foods-13-03000-t001:** Moisture content monitoring (% w.b.) during processes A and B. Process A is without ultrasonic application and process B includes two steps of ultrasonic applications. W is wet original BSG; P is pressed BSG; U1 is the first ultrasound application to BSG; U2 is the second ultrasound application to BSG; D is dried BSG; A1, A2, B1 and B2 are powders of BSG.

Process	W	U1	P	U2	D	A1 and B1	A2 and B2
A	72.4 ± 2.4 Aa	-	59.8 ± 1.6 Ab	-	5.2 ± 0.6 Ac	6.6 ± 0.5 Ac	6.1 ± 0.9 Ac
B	71.2 ± 1.7 Aa	72.1 ± 3.3 a	59.1 ± 1.1 Ab	60.4 ± 2.6 b	5.4 ± 3.1 Ac	6.4 ± 0.3 Ac	5.9 ± 0.1 Ac

BSG: wet original (W), first ultrasound (U1), pressed (P), second ultrasound (U2), dried (D) and powder (A1, A2, B1, B2) according to Figure 1. In each column, different capital letters indicate significant differences determined by the t-Student test at *p* < 0.05, while different lowercase letters in each row indicate significant differences based on Tukey’s test at *p* < 0.05. Data are presented as mean ± standard deviation.

**Table 2 foods-13-03000-t002:** Color characteristics of BSG along process A (without ultrasonic application) and B (including two steps of ultrasonic applications). W is wet original BSG; P is pressed BSG; U1 is the first ultrasound application to BSG (process B); U2 is the second ultrasound application to BSG (process B); D is dried BSG; A1 and B1, A2 and B2 are powders of BSG.

Color Parameter	Process	W	P	U1	U2	D	A1 and B1	A2 and B2
L*	A	44.1 ± 0.6 c	45.0 ± 1.1 c	-	-	48.7 ± 0.6 b	68.0 ± 1.4 a	63.4 ± 1.2 a
	B	44.4 ± 0.5 c	47.3 ± 1.3 b	44.8 ± 1.1 c	46.6 ± 1.0 c	49.9 ± 1.1 b	69.3 ± 2.1 a	66.5 ± 3.0 a
a*	A	5.3 ± 0.3 a	4.9 ± 0.3 a	-	-	3.4 ± 0.2 b	3.0 ± 0.3 b	3.4 ± 0.3 b
	B	5.4 ± 0.4 a	5.2 ± 0.4 a	5.5 ± 0.2 a	5.8 ± 0.4 a	3.7 ± 0.1 b	2.7 ± 0.6 c	3.0 ± 0.5 c
b*	A	22.5 ± 0.5 a	22.6 ± 0.8 a	-	-	21.5 ± 0.5 a	20.9 ± 0.2 a	21.4 ± 1.4 a
	B	22.8 ± 0.5 a	23.1 ± 0.3 a	22.8 ± 0.3 a	24.0 ± 0.4 a	21.8 ± 0.4 a	20.6 ± 0.5 a	21.2 ± 0.3 a
ΔE	A	-	1.3 ± 1.1 d	-	-	5.2 ± 1.0 c	24.3 ± 1.4 a	19.4 ± 1.6 b
	B	-	3.1 ± 1.4 d	1.1 ± 0.5 e	2.6 ± 0.8 d	5.9 ± 0.7 c	25.2 ± 2.2 a	22.4 ± 3.2 b
h*	A	81.8 ± 0.7 a	81.0 ± 0.3 a	-	-	76.8 ± 0.6 b	77.8 ± 0.7 b	81.0 ± 0.7 a
	B	82.5 ± 1.4 a	76.8 ± 0.6 c	82.0 ± 1.2 ab	76.6 ± 0.3 c	77.3 ± 0.9 c	76.4 ± 1.0 c	80.4 ± 0.5 b
C*	A	23.1± 0.4 a	23.1± 0.8 a	-	-	21.8 ± 0.5 a	21.1 ± 0.2 a	21.7 ± 1.7 a
	B	23.5 ± 0.6 a	23.7 ± 0.3 a	23.5 ± 0.3 a	24.7 ± 0.4 a	22.1 ± 0.4 a	20.8 ± 0.5 a	21.4 ± 0.3 a

BSG: wet original (W), first ultrasound (U1), pressed (P), second ultrasound (U2), dried (D) and powder (A1, A2, B1, B2) according to Figure 1. In each row, different letters indicate significant differences by Tukey’s test at *p* < 0.05. Data are presented as mean ± standard deviation.

**Table 3 foods-13-03000-t003:** ΔE values between the food powders obtained from the BSG in processes A and B. Process A is without ultrasonic application and process B includes two steps of ultrasonic applications. A1 and B1, A2 and B2 are powders of BSG.

	A2	B1	B2
A1	4.9 ± 1.0 ab	1.1 ± 1.0 d	1.8 ± 1.3 d
A2	-	5.7 ± 2.6 a	3.2 ± 1.9 b
B1	-	-	2.8 ± 1.4 bc

Different letters among ΔE values indicate significant differences based on Tukey’s test at *p* < 0.05. Data are presented as mean ± standard deviation.

**Table 4 foods-13-03000-t004:** TPC and ABTS•+ radical of BSG powders obtained by processes A and B. Process A is without ultrasonic application and process B includes two steps of ultrasonic applications. A1, A2, B1 and B2 are powders of BSG.

BSG Powder	TPC (mg GAE g^−1^ d.m.)	ABTS•+ (µmol TEAC g^−1^ d.m.)
A1	3.05 ± 0.28 a	30.58 ± 1.65 a
A2	3.04 ± 0.23 a	30.43 ± 1.98 a
B1	3.59 ± 0.43 a	24.72 ± 1.86 ab
B2	3.68 ± 0.33 a	22.81 ± 1.59 b

TPC: Total phenolic content. GAE: Gallic acid equivalent. TEAC: Trolox equivalent antioxidant capacity. d.m.: dry matter. In each column, different letters indicate significant differences based on Tukey’s test at *p* < 0.05. Data are presented as mean ± standard deviation.

**Table 5 foods-13-03000-t005:** Proximate composition, expressed in dry matter (% *w*/*w*), of BSG powders obtained by processes A and B. Process A is without ultrasonic application and process B includes two steps of ultrasonic applications. A1, A2, B1 and B2 are powders of BSG.

Component	A1	A2	B1	B2
Total lipid (fat)	8.7 ± 0.1 a	10.0 ± 0.6 a	8.6 ± 1.0 a	9.6 ± 0.0 a
Protein (N × 5.83)	18.1 ± 0.3 a	21.8 ± 1.9 a	21.5 ± 1.5 a	24.2 ± 2.1 a
Total dietary fiber	37.8 ± 0.7 c	45.5 ± 1.0 b	46.0 ± 0.2 b	56.1 ± 0.5 a
Soluble	4.0 ± 0.0 b	3.9 ± 0.4 b	4.9 ± 0.3 a	5.3 ± 0.1 a
Insoluble	33.8 ± 0.7 c	41.6 ± 1.4 b	41.1 ± 0.3 b	50.7 ± 0.6 a
Ash	2.3 ± 0.1 a	2.7 ± 0.2 a	2.2 ± 0.4 a	2.6 ± 0.4 a
Available carbohydrate (by difference)	33.5 ± 0.1 a	19.6 ± 1.7 b	21.8 ± 2.3 b	7.3 ± 1.7 c

In each row, different letters indicate significant differences by Tukey’s test at *p* < 0.05. Data are presented as mean ± standard deviation.

**Table 6 foods-13-03000-t006:** Powder particle sieve size distribution in percentage obtained by processes A and B. Process A is without ultrasonic application and process B includes two steps of ultrasonic applications. A1, A2, B1 and B2 are powders of BSG.

Particle Sieve Size (µm)	A1	A2	B1	B2
425	0.4 ± 0.0 a	0.3 ± 0.3 a	0.2 ± 0.2 a	0.3 ± 0.2 a
180	0.0 ± 0.0 a	0.5 ± 0.1 a	3.1 ± 2.8 a	1.4 ± 1.0 a
125	22.1 ± 1.1 b	21.3 ± 2.1 b	24.0 ± 0.3 ab	27.9 ± 2.1 a
90	35.4 ± 1.1 b	27.8 ± 0.2 c	38.3 ± 1.0 a	19.5 ± 1.5 d
75	36.3 ± 1.0 a	17.5 ± 2.1 d	30.1 ± 0.6 b	26.2 ± 1.5 c
45	4.6 ± 0.2 c	31.8 ± 0.5 a	3.8 ± 1.2 c	22.8 ± 0.8 b
<45	1.2 ± 0.8 a	0.8 ± 0.3 a	0.5 ± 0.2 a	2.0 ± 1.1 a

In each row, different letters indicate significant differences based on Tukey’s test at *p* < 0.05. Data are presented as mean ± standard deviation.

## Data Availability

The original contributions presented in the study are included in the article, further inquiries can be directed to the corresponding authors.

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
