# Peer review of "Effect of Physical Separation with Ultrasound Application on Brewers’ Spent Grain to Obtain Powders for Potential Application in Foodstuffs"

_foods, 2024, doi:10.3390/foods13183000_

Round 1

Reviewer 1 Report

Comments and Suggestions for Authors

This manuscript is dealing with the ultrasound application on brewers’ spent grain. My comments about the manuscript are as following:

1. The keywords "flour" should be reconsidered. It hardly seems to appear in the manuscript.

2. Pay attention to the section number, e.g. Line 111: "3. Material and methods".

3. Table 1, what are the meanings of "Powder 1" and "Powder 2"

4. If the data of wet original (W) in Table 2 and 3 is different, how could the authors compare the Process A and B?

5. Line 584: which kind of " antioxidant substances"?

6. In conclusions, what are the advantages of the ontained powders used in  plant-based foodstuffs?

Author Response

Temuco, September 6, 2024

Manuscript Number: Foods 3140595 

Point-by-point response to reviewers’ comments

Dear Editor and Reviewers,

We appreciated the comments and suggestions of the referees, which clearly improved the manuscript. They were all addressed in the revised version, and the necessary changes and amendments were made, and outlined in red.

We hope that now the manuscript complies with the high-quality requirements for publication in Foods.

Sincerely yours,

Erick Sigisfredo Scheuermann Salinas

Chemical Engineering Department

Center of Food Biotechnology and Bioseparations (BIOREN) and Biotechnological Research Center Applied to the Environment (CIBAMA)

Universidad de La Frontera

Severino Matias de Alencar

Department of Food Science and Technology

Escola Superior de Agricultura Luiz Queiroz (ESALQ)

Universidade de São Paulo

Reviewer 1

Open Review

( ) I would not like to sign my review report

(x) I would like to sign my review report

Quality of English Language

( ) I am not qualified to assess the quality of English in this paper.

( ) The English is very difficult to understand/incomprehensible.

( ) Extensive editing of English language required.

( ) Moderate editing of English language required.

( ) Minor editing of English language required.

(x) English language fine. No issues detected.

Yes    Can be improved     Must be improved    Not applicable

Does the introduction provide sufficient background and include all relevant references?

(x)      ( )       ( )       ( )

Is the research design appropriate?

(x)      ( )       ( )       ( )

Are the methods adequately described?

( )       (x)      ( )       ( )

Are the results clearly presented?

( )       (x)      ( )       ( )

Are the conclusions supported by the results?

( )       (x)      ( )       ( )

Comments and Suggestions for Authors

This manuscript is dealing with the ultrasound application on brewers’ spent grain. My comments about the manuscript are as following:

  1. The keywords "flour" should be reconsidered. It hardly seems to appear in the manuscript.

Thank you for your recommendation. The word “flour” was removed, and the word “powder” was included.

  1. Pay attention to the section number, e.g. Line 111: "3. Material and methods".

Thank you for your recommendation. Removed the numbering of numbered sections and differentiated sections by font size and italics.

  1. Table 1, what are the meanings of "Powder 1" and "Powder 2"

Thank you for your question, which made us clarify it in the revised version. Powder 1 and Powder 2 were replaced by A1 and B1, A2 and B2, which are the identifications that appear in Figure 1.

  1. If the data of wet original (W) in Table 2 and 3 is different, how could the authors compare the Process A and B?

Thank you for your question. The moisture content data of the original BSG (W) are found in Table 1, and it is reported that there is no statistically significant difference between the BSG used in Process A (72.4 ± 2.4) and that used in Process B (71.2 ± 1.7). Therefore, the color analysis shown in Tables 2 and 3 contemplate the use of samples whose moisture contents do not differ from each other.

  1. Line 584: which kind of " antioxidant substances"?

Thank you for your question. It was included: such as phenolic compounds.

One of the most important classes of antioxidants found in vegetable sources is the phenolic compounds. In the specific case of BSG, phenolic acids and flavan-3-ols are responsible for its antioxidant activity. We added this information in the revised version of the manuscript.

Naibaho, J., Wojdyło, A., Korzeniowska, M., Laaksonen, O.; Föste, M., Kütt, M.L., Yang, B. Antioxidant activities and polyphenolic identification by UPLC-MS/MS of autoclaved brewers’ spent grain. LWT, Volume 163, 2022, 113612, ISSN 0023-6438, https://doi.org/10.1016/j.lwt.2022.113612.

  1. In conclusions, what are the advantages of the contained powders used in plant-based foodstuffs?

Thank you for your question. We included the following phase in the revised version of the manuscript: “Overall, the powders derived from brewers' spent grains have a high content of dietary soluble fiber, making them valuable for enriching food products. Additionally, they contribute protein content and antioxidant activity. Regarding protein content, the powders obtained from both Processes A and B may be suitable for use in various plant-based foodstuffs, which is particularly relevant given the growing vegetarian and vegan population.”

Reviewer 2

Open Review

(x) I would not like to sign my review report

( ) I would like to sign my review report

Quality of English Language

( ) I am not qualified to assess the quality of English in this paper.

( ) The English is very difficult to understand/incomprehensible.

(x) Extensive editing of English language required.

( ) Moderate editing of English language required.

( ) Minor editing of English language required.

( ) English language fine. No issues detected.

Yes    Can be improved     Must be improved    Not applicable

Does the introduction provide sufficient background and include all relevant references?

( )       ( )       (x)      ( )

Is the research design appropriate?

( )       ( )       (x)      ( )

Are the methods adequately described?

( )       ( )       (x)      ( )

Are the results clearly presented?

( )       ( )       (x)      ( )

Are the conclusions supported by the results?

( )       ( )       (x)      ( )

Comments and Suggestions for Authors

The work deals with the search for the valorization of waste from the brewing industry. Separation, ultrasound, drying and milling processes are applied.

The work presents some errors of concept, severe problems of organization, drafting and thoroughness. Here are some opportunities for improvement.

It is necessary to improve the writing, since it is a difficult to read document, with errors in the English wording and some redundant parts, it seems to be a text coming from an undergraduate work.

The title could be improved, avoid phrases like with or without ultrasound, but rather talk about the effect of ultrasound application.

Thank you for your recommendation, which clearly helped to improve the title. A new title is: “Effect of physical separation with ultrasound application on brewers’ spent grain to obtain powders for potential application in foodstuffs”

It is not necessary to include orcid in this part l4-l8.

The ORCID numbers were deleted.

The abstract should be reworded

Thank you for your recommendation. The abstract has been reworded and the English was revised by a native speaker, according to the reviewer's suggestion, aiming to improve the clarity and conciseness of the text.

L38 its application depends on processing as more general, rather than drying.

Thank you for your recommendation.  It was change in the abstract.

In l 39 evaluate the effect of ultrasound.... eliminate with or without

Thank you for your recommendation. The new text is: …effect of physical separation with ultrasound application on color….

In l40 wet bsg is mentioned, it should not be fresh bsg.

Thank you for your comment. We agree that fresh characterizes better the BSG, however, the term wet BSG is well stablished in the literature, as it is shown below in examples, and this term will be more likely used in searches:

Lynch, K.M.; Steffen, E.J.; Arendt, E.K. Brewers’ spent grain: a review with an emphasis on food and health. J. Inst. Brew. 2016, 122(4), 553–568. https://doi.org/10.1002/jib.36

Pabbathi, A.; Velidandi, S.; Pogula, P.K.; Gandam, R.R.; Baadhe, M.; Sharma, R.; Sirohi, V. K.; Thakur, V.K.; Gupta, V.K. Brewer's spent grains-based biorefineries: a critical review. Fuel 2022, 317, 123435 (). https://doi.org/10.1016/j.fuel.2022.123435

Ibbett, R.; White, R.; Tucker, G.; Foster, T. Hydro-mechanical processing of brewer's spent grain as a novel route for separation of protein products with differentiated techno-functional properties. Innov. Food Sci. Emerg. Techno. 2019, 56, 102184. https://doi.org/10.1016/j.ifset.2019.102184

Garrett, R.; Bellmer, D.; McGlynn, W.; Rayas-Duarte, P. Development of new chip products from brewer's spent grain. J. Food Qual. 2021, 2021, 5521746. https://doi.org/10.1155/2021/5521746

Santos, M.V.; Ranalli, N.; Orjuela-Palacio, J.; Zaritzky, N. Brewers spent grain drying: drying kinetics, moisture sorption isotherms, bioactive compounds stability, and Bacillus cereus lethality during thermal treatment. J. Food Eng. 2024, 364, 111796. https://doi.org/10.1016/j.jfoodeng.2023.111796

l66 refers to global production, does it mean worldwide production?

Thank you for your question. The English was revised by a native speaker, and this sentence was changed to “It is estimated that more than 180 million tons of BSG are produced globally each year, accounting for 85% of brewery residues, with most of it being discarded or used as low-value animal feed [2]”.

l72-72 Revise the sentence about the amount of energy needed for microbiological stabilization?

Thank you for your recommendation. The English was revised by a native speaker, and this sentence was changed to “However, with a moisture content of 70 to 80%, BSG is prone to rapid microbiological deterioration. Drying is typically employed to stabilize BSG by reducing its moisture level to around 6%, but this process demands considerable energy [3, 5, 6].” 

The paragraph at the end of l73 seems a bit out of place, maybe you could expand it to better expose this idea.

Thank you for your recommendation. The English was revised by a native speaker, and this sentence was changed to “To mitigate the energy consumption associated with drying, pre-pressing the BSG to lower its initial moisture content has been suggested [7].”

In l80 it talks about lignocellulosic biomass, it refers to BSG, review and standardize, the idea about the potential at large scales is not clear either.

Thank you for your recommendation. The English was revised by a native speaker, and this sentence was changed to “Given that BSG is a lignocellulosic by-product, high-power ultrasound can be particularly effective in breaking down cell structures, thereby facilitating the release of nutrients and other compounds contained within the grains. Ultrasound technology, therefore, holds potential for large-scale processes involving the pretreatment of BSG’s lignocellulosic biomass [9].”

l89-90 The idea should be presented in more detail, as it is, it looks like it is not clear from the rest of the text...

Thank you for your recommendation. The English was revised by a native speaker, and this sentence was changed to “Ultrasound has also been employed in combination with convective drying to enhance moisture migration and improve the quality of dried products. Applied prior to drying, ultrasound not only improves moisture removal but also enhances the structural integrity, chemical composition, texture, and retention of bioactive compounds [14–16].”

L92 What is the meaning of talking about digestibility?

Thank you for your careful reading and question. The English was revised by a native speaker, and we changed the text to make it clearer, as follows: “Milling, a critical unit operation, reduces the molecular size of polysaccharides by disrupting cell structures and decreasing cellulose crystallinity. This reduction in particle size facilitates the hydrolysis of cellulose, hemicellulose, and lignin, while also decreasing the degree of polymerization of these molecules [17]. Additionally, milling can increase the exposure of proteins within the matrix to proteases during digestion, potentially improving protein digestibility. Further research in this area is warranted.”

l98-l102 Review relevance

We consider it relevant because it is through sieving that the physical separation that contributes to the different characteristics of the powders is achieved.

Revise the wording of the objective of the work in l106-109.

Thank you for your recommendation. It was eliminated “or without”, the English was revised by a native speaker, and the text was changed to “The objective of this study was to evaluate the effects of physical separation with ultrasound application on the instrumental color, total phenolic content, antioxidant activity, proximate composition, total dietary fibers, soluble and insoluble fibers, and particle size distribution of BSG powders for potential use in food products.”

In l116, what is the reason for 1,010g would it suffice to say 1 kilo?

Thank you for your question. The following text was included in the revised version of the manuscript: “The BSG was portioned into 1.010 g samples, packaged in Ziploc® bags (26.8 x 27.3 cm, USA), and immediately frozen at -18 °C. The additional 10 grams in each sample were allocated for moisture measurement in each of the experimental replicates after the BSG was thawed. This step was taken to ensure the accuracy of the measurements required for the study.”

l118, delete the last sentence that does not make sense.

Thank you for your recommendation. It was deleted.

Check if it is possible to join the two processes in one diagram.

Thank you for your recommendation. We merged the two processes in one diagram for Figure 1.

l125-l128 revise the wording.

Thank you for your recommendation. the English was revised by a native speaker, and “Described in 3.3.1. and 3.3.2.” was replaced with “described in the section "Characterization of the BSG Powders".

l134-l136 is it really necessary?

Thank you for your question. It was deleted and some text is included in Fig. 1: “Sequences of Operations for Processes A and B to Obtain Food Powders from BSG and Sampling Points. Letters in red indicate the processes within the boxes below them.”

l152 is it necessary to talk about unmarked equipment?

Thank you for your question. “(equipment without manufacturer's brand)” was deleted.

l154 how were the liquid fractions stored?

Thank you for your question. Text was added to clarify it: “The liquid fraction was stored in Schott glass bottles and frozen at -18 °C for preservation, while the solid fraction was subjected to further processing to obtain powders.”

l155 any formula for the calculation of the yields?

Thank you for your question. Text was added to clarify it: “The yield of the pressing operation, in terms of liquid fraction, was determined by dividing the volume of the liquid fraction by the initial mass of BSG.”

l155 "de" powders refers to the powders? review

Thank you for your careful reading. The necessary amendments were made.

It is not clear how the drying process was developed, i.e. if there is air circulation and its speed in the equipment.

Thank you for your question. Text was added to clarify it: “…with air circulation at nominal velocity of 2.5 m s-1.”

l161 owner, did you mean oven?

Thank you for your careful reading. The necessary amendments were made.

In the sieving process, did you calculate the average particle diameter?

Thank you for your question. No. We only determined the particle size distribution by sieving, since it was the equipment available for this characterization, and we believe it is adequate to achieve the objective of this paper, since industrially the particle sizes of powders are characterized generally by this sieving (sifting) approach.

l183 vapor tight aluminuim bags?

Thank you for your careful reading. We excluded “tight”.

In the color measurements, it is necessary to specify the illuminant and observer used, as well as the thickness of the samples.

Thank you for your recommendation. Text was added to clarify it: “…CIE condition C (6774 K) illuminant, CIE 1931 2° observer, with a sample thickness of 0.5 cm.”

Why in some analyses it measures in sextuplicate and in others only in duplicate, it is necessary to justify

Thank you for your question. Further explanation was added to the text: “Data for each process were collected in sextuplicate. Color measurements were also conducted in sextuplicate, as described in the "Instrumental Color" section. Total phenolic content, antioxidant activity against the ABTS•+ radical, total dietary fiber, soluble and insoluble fiber, and particle size distribution were measured in triplicate. Proximate composition was performed in duplicate due to sample quantity limitations and the material required for this analysis.”

In 4.1.1 it talks about monitoring, they are only measurements... what is the sense?

Thank you for your question. To clarify it, we included the following text: “Moisture content and instrumental color were monitored throughout Processes A and B to assess any changes in the BSG attributed to the various operations.”

l307 Was the sacharification measured?

Thank you for your comment. We removed this sentence in order to avoid confusion, since we just wanted to explain how it is measured, but we did not perform this analysis.

l327 Did you measure the structural modifications? permeability of the bags?

Thank you for your question. No structural modifications of the BSG or the permeability of the packaging were measured, it is only a statement of the possible cause of the increase in moisture content. This would have to be evaluated in the future.

l322 - l334 If something was not done it is not necessary to write it down, otherwise it is not necessary to have a scale connected to make drying kinetics.

Thank you for your question. The text was deleted.

l344-345 what is the point of comparing with cabbage?*

Thank you for your question. The point of comparison is the size of the particles, which significantly influences moisture removal; the larger the particles, the more difficult it is to extract moisture. This is because large particles have a smaller surface area relative to their volume.

l390, 393 revise the sentence

Thank you for your recommendation. The text was revised by a native speaker, and changes were made: “The color changes observed in BSG during drying (D) using hot-air methods are likely due to the Maillard reaction and caramelization, which tend to produce a darker product [12]. Additionally, the Maillard reactions in BSG have been documented during extrusion grinding via thermo-mechanical treatment in a twin-screw extruder [38, 39].”

Tables 2 and 3 can be joined

Thank you for your suggestion. We believe that joining Tables 2 and 3 would not allow a clear visualization of the data and discussion of the results, so we consider it more appropriate to keep both tables.

The analysis of the results is somewhat poor and is only limited to presenting the data obtained without a deeper analysis in the specific context of each case.

We believe that, with your comments and the incorporation of explanations, it has been possible to enrich the analysis of results in this manuscript.

What is the contribution of figure 2, it is necessary to justify its relevance?

Thank you for your question. Text was added to justify it: “These graphs are crucial for determining whether the BSG was exposed to temperatures that could induce biochemical changes during ultrasound treatment.”

The conclusion should be rewritten with emphasis on the knowledge generated and its possible applications.

Thank you for your recommendation. Conclusion section was rewritten, and the English was revised by a native speaker.

Comments on the Quality of English Language

There are important wording problems and the quality of the English is not good, it is strongly recommended that it be revised.

English was revised by a native speaker throughout the text.

Reviewer 2 Report

Comments and Suggestions for Authors

The work deals with the search for the valorization of waste from the brewing industry. Separation, ultrasound, drying and milling processes are applied.

The work presents some errors of concept, severe problems of organization, drafting and thoroughness. Here are some opportunities for improvement.

It is necessary to improve the writing, since it is a difficult to read document, with errors in the English wording and some redundant parts, it seems to be a text coming from an undergraduate work.

The title could be improved, avoid phrases like with or without ultrasound, but rather talk about the effect of ultrasound application.

It is not necessary to include orcid in this part l4-l8.

The abstract should be reworded

L38 its application depends on processing as more general, rather than drying.

In l 39 evaluate the effect of ultrasound.... eliminate with or without

In l40 wet bsg is mentioned, it should not be fresh bsg.

l66 refers to global production, does it mean worldwide production?

l72-72 Revise the sentence about the amount of energy needed for microbiological stabilization?

The paragraph at the end of l73 seems a bit out of place, maybe you could expand it to better expose this idea.

In l80 it talks about lignocellulosic biomass, it refers to BSG, review and standardize, the idea about the potential at large scales is not clear either.

l89-90 The idea should be presented in more detail, as it is, it looks like it is not clear from the rest of the text...

L92 What is the meaning of talking about digestibility?

l98-l102 Review relevance

Revise the wording of the objective of the work in l106-109.

In l116, what is the reason for 1,010g would it suffice to say 1 kilo? 

l118, delete the last sentence that does not make sense.

Check if it is possible to join the two processes in one diagram.

l125-l128 revise the wording.

l134-l136 is it really necessary?

l152 is it necessary to talk about unmarked equipment?

l154 how were the liquid fractions stored?

l155 any formula for the calculation of the yields?

l155 "de" powders refers to the powders? review

It is not clear how the drying process was developed, i.e. if there is air circulation and its speed in the equipment.

l161 owner, did you mean oven?

In the sieving process, did you calculate the average particle diameter?

l183 vapor tight aluminuim bags?

In the color measurements, it is necessary to specify the illuminant and observer used, as well as the thickness of the samples.

Why in some analyses it measures in sextuplicate and in others only in duplicate, it is necessary to justify

In 4.1.1 it talks about monitoring, they are only measurements... what is the sense?

l307 Was the sacharification measured?

l327 Did you measure the structural modifications? permeability of the bags?

l322 - l334 If something was not done it is not necessary to write it down, otherwise it is not necessary to have a scale connected to make drying kinetics.

l344-345 what is the point of comparing with cabbage?

l390, 393 revise the sentence

Tables 2 and 3 can be joined

The analysis of the results is somewhat poor and is only limited to presenting the data obtained without a deeper analysis in the specific context of each case.

What is the contribution of figure 2, it is necessary to justify its relevance?

The conclusion should be rewritten with emphasis on the knowledge generated and its possible applications.

Comments on the Quality of English Language

There are important wording problems and the quality of the English is not good, it is strongly recommended that it be revised.

Author Response

(The authors gave the same response as above.)

Round 2

Reviewer 1 Report

Comments and Suggestions for Authors

I suggest this paper can be accepted.

Author Response

Temuco, September 19, 2024

Manuscript Number: Foods 3140595 

Point-by-point response to Academic Editor comments

Dear Academic Editor,

We appreciated the last comment and suggestion of the Academic Editor, which clearly improved the manuscript. They were all addressed in the revised version, and the necessary changes were made, and outlined in blue.

We hope that now the manuscript complies with the high-quality requirements for publication in Foods.

Sincerely yours,

Erick Sigisfredo Scheuermann Salinas

Chemical Engineering Department

Center of Food Biotechnology and Bioseparations (BIOREN) and Biotechnological Research Center Applied to the Environment (CIBAMA)

Universidad de La Frontera

Severino Matias de Alencar

Department of Food Science and Technology

Escola Superior de Agricultura Luiz Queiroz (ESALQ)

Universidade de São Paulo

Please, avoid the use of the 1st person in the manuscript (we propose, we analyzed, we believe, we used, we considered) and use impersonal or passive voice instead (it is proposed, it was analyzed, etc.)

Thank you, the changes to impersonal or passive voice was made.

Table captions should be more self-explanatory and describe all the abbreviations used in the table or in the caption itself. The same for figure captions. For instance:

- Figure 1: Processes A and B should be detailed/explained in the caption.

- Table 1: Processes A/B should be detailed. Besides, abbreviations W, U1, P, etc. need to be explained in the caption.

Thank you. For all Figure and Tables, detailed/explained of Processes A/B and abbreviations W, U1, P, etc.  were includes in captions.

- Table 2. “Color monitoring concerning the wet original BSG for process A”. More than color monitoring, it should say “color characteristics of BSG along process A (specify).” Please, include abbreviations and explanations for W, P, D, etc. As far as I understand, only W refers to “wet original BSG”.

Thanks you. The changes were made.

- Table 3. The same than for Table 2. I suggest combining Table 2 and 3 in a single table.

Tables 2 and 3 were combining. Then, the changes were made in the numbering of Tables 4 to 7, for Tables 3 to 6.

- Please extend the previous recommendations to all Figure and Table captions and make the corresponding changes.

For all Figure and Tables, detailed/explained of Processes A/B and abbreviations W, U1, P, etc.  were includes in captions.

I believe the conclusions of this work could be more addressed. I suggest the authors to be more specific in paragraph 1/conclusions, when it is said that the different processes yielded different products with different characteristics, but no differences are specifically cited. Please try to highlight a main result/trend which might determine the application, for instance. When referring to process A and B in the conclusions, it is better that the reader might understand what are the differences between A and B without reading the whole manuscript (note that sometimes abstract and conclusions are read before deciding to read the whole manuscript). Please revise grammar and spelling in the conclusion (line 815 “to protein content”)

Thank you. The suggestions were attended.

Reviewer 2 Report

Comments and Suggestions for Authors

The work improved considerably in terms of its clarity and contribution to the discipline.

I would just like to raise a question about the contribution of figure 2, could U1 and U2 be united in the same graph?

Author Response

Temuco, September 7, 2024

Manuscript Number: Foods 3140595 

Point-by-point response to reviewers’ comments

We appreciated the last comment and suggestion of the reviewer, which clearly improved the manuscript. They were all addressed in the revised version, and the necessary changes were made, and outlined in green.

We hope that now the manuscript complies with the high-quality requirements for publication in Foods.

Sincerely yours,

Erick Sigisfredo Scheuermann Salinas

Chemical Engineering Department

Center of Food Biotechnology and Bioseparations (BIOREN) and Biotechnological Research Center Applied to the Environment (CIBAMA)

Universidad de La Frontera

Severino Matias de Alencar

Department of Food Science and Technology

Escola Superior de Agricultura Luiz Queiroz (ESALQ)

Universidade de São Paulo

Open Review

(x) I would not like to sign my review report

( ) I would like to sign my review report

Quality of English Language

(x) I am not qualified to assess the quality of English in this paper.

( ) The English is very difficult to understand/incomprehensible.

( ) Extensive editing of English language required.

( ) Moderate editing of English language required.

( ) Minor editing of English language required.

( ) English language fine. No issues detected.

Yes        Can be improved          Must be improved       Not applicable

Does the introduction provide sufficient background and include all relevant references?

(x)          ( )          ( )          ( )

Is the research design appropriate?

(x)          ( )          ( )          ( )

Are the methods adequately described?

(x)          ( )          ( )          ( )

Are the results clearly presented?

(x)          ( )          ( )          ( )

Are the conclusions supported by the results?

(x)          ( )          ( )          ( )

Comments and Suggestions for Authors

The work improved considerably in terms of its clarity and contribution to the discipline.

I would just like to raise a question about the contribution of figure 2, could U1 and U2 be united in the same graph?

Thank you for your recommendation. We united the temperature-time curves of the U1 and U2 ultrasound operation in the same graph (Fig 2), and to delete the Figure 3. Then the Figure 4 was named as Figure 3. Changes are in lines 469, 482, 490-491, 509, 520 and 525 with letters in green color.
